# Establishing an online resource to facilitate global collaboration and inclusion of underrepresented populations: Experience from the MJFF Global Genetic Parkinson's Disease Project

Eva-Juliane Vollstedt[1], Harutyun Madoev[1], Anna Aasly[2], Azlina Ahmad-Annuar[3], Bashayer Al-Mubarak[4], Roy N. Alcalay[5,6], Victoria Alvarez[7,8], Ignacio Amorin[9], Grazia Annesi[10], David Arkadir[11], Soraya Bardien[12,13], Roger A. Barker[14], Melinda Barkhuizen[15], A. Nazli Basak[16], Vincenzo Bonifati[17], Agnita Boon[18], Laura Brighina[19], Kathrin Brockmann[20,21], Andrea Carmine Belin[22], Jonathan Carr[13,23], Jordi Clarimon[24,25], Mario Cornejo-Olivas[26,27], Leonor Correia Guedes[28,29], Jean-Christophe Corvol[30], David Crosiers[31,32,33], Joana Damásio[34,35], Parimal Das[36], Patricia de Carvalho Aguiar[37,38], Anna De Rosa[39], Jolanta Dorszewska[40], Sibel Ertan[41], Rosangela Ferese[42], Joaquim Ferreira[32,43], Emilia Gatto[44], Gençer Genç[45], Nir Giladi[6], Pilar Gómez-Garre[46,47], Hasmet Hanagasi[48], Nobutaka Hattori[49], Faycal Hentati[50], Dorota Hoffman-Zacharska[51], Sergey N. Illarioshkin[52], Joseph Jankovic[53], Silvia Jesús[46,47], Valtteri Kaasinen[54,55,56], Anneke Kievit[17], Peter Klivenyi[57], Vladimir Kostic[58], Dariusz Koziorowski[59], Andrea A. Kühn[60], Anthony E. Lang[61], Shen-Yang Lim[62], Chin-Hsien Lin[63], Katja Lohmann[1], Vladana Markovic[58], Mika Henrik Martikainen[54,56,64], George Mellick[65], Marcelo Merello[64,66,67], Lukasz Milanowski[59], Pablo Mir[46,47], Özgür Öztop-Çakmak[41], Márcia Mattos Gonçalves Pimentel[68], Teeratorn Pulkes[69], Andreas Puschmann[70,71], Ekaterina Rogaeva[72], Esther M. Sammler[73,74], Maria Skaalum Petersen[75,76], Matej Skorvanek[77,78], Mariana Spitz[79], Oksana Suchowersky[80], Ai Huey Tan[62], Pichet Termsarasab[69], Avner Thaler[6], Vitor Tumas[81], Enza Maria Valente[82,83], Bart van de Warrenburg[84], Caroline H. Williams-Gray[14], Ruey-Mei Wu[63], Baorong Zhang[85], Alexander Zimprich[86], Justin Solle[87], Shalini Padmanabhan[87], Christine Klein[1]*

1 Institute of Neurogenetics, University of Luebeck, Luebeck, Germany, 2 Department of Neuromedicine and Movement Science, Norwegian University of Science and Technology, Trondheim, Norway, 3 Department of Biomedical Science, Faculty of Medicine, University of Malaya, Kuala Lumpur, Malaysia, 4 Center for Genomic Medicine, Research Centre, King Faisal Specialist Hospital and Research Center, Riyadh, Saudi Arabia, 5 Department of Neurology, Columbia University, New York, New York, United States of America, 6 Neurological Institute, Tel-Aviv Medical Center, Tel-Aviv, Israel; Sackler School of Medicine, Sagol School of Neuroscience, Tel-Aviv University, Tel-Aviv, Israel, 7 Laboratório de Genética, Hospital Universitario Central de Asturias, Oviedo, Asturias, Spain, 8 Instituto de Investigación Sanitaria del Principado de Asturias (ISPA), Oviedo, Spain, 9 Universidad de la Republica Uruguay, Montevideo, Uruguay, 10 Institute of Biomedical Research and Innovation, National Research Council, Cosenza, Italy, 11 Department of Neurology, Hadassah Medical Center and the Hebrew University, Jerusalem, Israel, 12 Division of Molecular Biology and Human Genetics, Faculty of Medicine and Health Sciences, Stellenbosch University, Cape Town, South Africa, 13 South African Medical Research Council/Stellenbosch University Genomics of Brain Disorders Research Unit, Stellenbosch University, Cape Town, South Africa, 14 Department of Clinical Neurosciences, University of Cambridge, Cambridge, United Kingdom, 15 DST/NWU Preclinical Drug Development Platform, North-West University, Potchefstroom, North-West, South Africa, 16 Suna and Inan Kiraç Foundation, Neurodegeneration Research Laboratory, KUTTAM, School of Medicine, Koç University, Istanbul, Turkey, 17 Department of Clinical Genetics, Erasmus MC University Medical Center Rotterdam, Rotterdam, The Netherlands, 18 Department of Neurology, Erasmus MC, University Medical Center Rotterdam, Rotterdam, Netherlands, 19 Department of Neurology, Milan Center for Neuroscience, University of Milano-Bicocca/San Gerardo Hospital, Monza, Monza Brianza, Italy, 20 Department of Neurodegenerative Diseases, University of Tuebingen, Tuebingen, Baden Wuerttemberg, Germany, 21 Hertie Institute for Clinical Brain Research and German Centre for Neurodegenerative Diseases, Tuebingen, Baden



**Data Availability Statement:** The data underlying the results presented in the study are available

from https://gp2networkdev.wpengine.com/monogenic-resource-map/.

**Funding:** CK received a grant for this project by the Michael J Fox Foundation (ID 15015.03, https://www.michaeljfox.org/). Roger Barker and Caroline Williams-Gray are supported by the NIHR Cambridge Biomedical Research Centre (BRC-1215-20014). The views expressed are those of the authors and not necessarily those of the NIHR or the Department of Health and Social Care. The funders reviewed the study design and suggested additional items for the survey. They had no role in data collection and analysis, the decision to publish, or the preparation of the manuscript.

**Competing interests:** I have read the journal's policy and the authors of this manuscript have the following competing interests: Roy N. Alcalay received consulting fees from Avrobio, Caraway, Ono Therapeutics, GSK, Merck, Sanofi, Janssen and grants from the Michael J. Fox Foundation, DOD, the Parkinson's Disease Foundation, and the NIH. Melinda Barkhuizen received a PhD scholarship from the National Research Foundation of South Africa (grant numbers 89230 and 98217) and internal funding from the research center where the study was conducted (DST/NWU Preclinical Drug Development Platform, North-West University, South Africa); Ampath Pathology laboratories in South Africa donated services in the form of drawing participant blood for DNA extractions, Several neurologists in South Africa assisted the authors with identifying patients with Parkinson's disease and referring them to the genotyping study; the North-West University, South Africa provided Ethical oversight and approval of the genotyping project. Vincenzo Bonifati received grants from Stichting Parkinson Fonds (Netherlands) and Alzheimer Nederland, he received honoraria from the International Parkinson and Movement Disorder Society (MDS), for lectures and as Chair of the International Congress Scientific Program Committee (2020-2021), and from Elsevier Ltd. as co-Editor in Chief of the journal Parkinsonism & Related Disorders (2018-current); he is unpaid member of the Stichting Parkinson Fonds (Netherlands). Kathrin Brockmann received grants from the German Federal Ministry of Education and Research (BMBF; PDStrat; FKZ 031L0137B) and from the German Center for Neurodegenerative Diseases (DZNE), consulting fees from Abbvie, Lundbeck, UCB, Zambon, Roche, and honoraria from Abbvie and UCB. Jordi Clarimon is full-time employee at Lundbeck A/S (Denmark). Mario Cornejo-Olivas has subcontracts with Cleveland Clinic and San Marcos Foundation for recruiting participants for

Wuerttemberg, Germany, **22** Department of Neuroscience, Karolinska Institutet, Stockholm, Sweden, **23** Division of Neurology, Department of Medicine, Faculty of Medicine and Health Sciences, Stellenbosch University, Cape Town, South Africa, **24** Department of Neurology, Biomedical Research Institute IIB-Sant Pau, Hospital de la Santa Creu i Sant Pau, Barcelona, Spain, **25** Centro de Investigación Biomédica en Red sobre Enfermedades Neurodegenerativas (CIBERNED), Madrid, Spain, **26** Neurogenetics Research Center, Instituto Nacional de Ciencias Neurologicas, Lima, Peru, **27** Center for Global Health, Universidad Peruana Cayetano Heredia, Lima, Peru, **28** Centro de Estudos Egas Moniz, Faculdade de Medicina, Universidade de Lisboa, Lisboa, Portugal, **29** Instituto de Medicina Molecular João Lobo Antunes, Faculty of Medicine, University of Lisbon, Lisbon, Portugal, **30** Paris Brain Institute—ICM, Inserm, CNRS, Assistance Publique Hôpitaux de Paris, Pitié-Salpêtrière Hospital, Department of Neurology, Sorbonne University, Paris, France, **31** Department of Neurology, Antwerp University Hospital, Edegem, Belgium, **32** Translational Neurosciences, Born Bunge Institute, Faculty of Medicine and Health Sciences, University of Antwerp, Wilrijk, Antwerp, Belgium, **33** Center for Molecular Neurology, VIB, Wilrijk, Antwerp, Belgium, **34** Department of Neurology, Hospital de Santo António—Centro Hospitalar Universitário do Porto, Porto, Portugal, **35** UnIGENe, Instituto de Biologia Molecular e Celular (IBMC), Instituto de Investigação e Inovação em Saúde (i3S), Universidade do Porto, Porto, Portugal, **36** Centre for Genetic Disorders, Institute of Science, Banaras Hindu University, Varanasi, Uttar Pradesh, India, **37** Hospital Israelita Albert Einstein, São Paulo, Brazil, **38** Department of Neurology and Neurosurgery, Universidade Federal de São Paulo, São Paulo, Brazil, **39** Department of Neurosciences and Reproductive and Odontostomatological Sciences, Federico II University, Naples, Italy, **40** Laboratory of Neurobiology, Department of Neurology, Poznan University of Medical Sciences, Poznan, Poland, **41** Department of Neurology, School of Medicine, Koç University, Istanbul, Turkey, **42** IRCCS Neuromed, Localita' Camerelle, Pozzilli, Isernia, Italy, **43** Laboratory of Clinical Pharmacology and Therapeutics, University of Lisbon, Lisbon, Portugal, **44** Movement Disorders, Department of Neurology, Instituto de Neurosciencias Buenos Aires, Buenos Aires, Argentina, **45** Department of Neurology, University of Health Sciences, Şişli Hamidiye Etfal Training and Research Hospital, İstanbul, Turkey, **46** Unidad de Trastornos del Movimiento, Servicio de Neurología y Neurofisiología Clínica, Instituto de Biomedicina de Sevilla, Hospital Universitario Virgen del Rocío/CSIC/Universidad de Sevilla, Seville, Spain, **47** Centro de Investigación Biomédica en Red sobre Enfermedades Neurodegenerativas (CIBERNED), Madrid, Spain, **48** Department of Neurology, Istanbul Faculty of Medicine, Istanbul University, Istanbul, Turkey, **49** Department of Neurology, Juntendo University School of Medicine, Bunkyo, Tokyo, Japan, **50** Mongi Ben Hmida National Institute of Neurology, Tunis, Tunisia, **51** Institute of Genetics and Biotechnology, Faculty of Biology, University of Warsaw, Warsaw, Poland, **52** Department of Neurogenetics, Research Center of Neurology, Moscow, Russia, **53** Parkinson's Disease Center and Movement Disorders Clinic, Department of Neurology, Baylor College of Medicine, Houston, Texas, United States of America, **54** Neurocenter, Turku University Hospital, Turku, Finland, **55** Department of Neurology, Satasairaala Hospital, Pori, Finland, **56** Clinical Neurosciences, Faculty of Medicine, University of Turku, Turku, Finland, **57** Department of Neurology, University of Szeged, Szeged, Hungary, **58** Department for Neurodegeneration, Clinic for Neurology UCCS, Medical Faculty, University of Belgrade, Belgrade, Serbia, **59** Department of Neurology, Faculty of Health Science, Medical University in Warsaw, Warsaw, Poland, **60** Movement Disorder and Neuromodulation Unit, Charité, Department of Neurology, Campus Mitte, Universitätsmedizin Berlin, Berlin, Germany, **61** Edmond J. Safra Program in Parkinson's Disease, Division of Neurology, Department of Medicine, Toronto Western Hospital, University of Toronto, Toronto, Ontario, Canada, **62** Division of Neurology, Department of Medicine, and the Mah Pooi Soo & Tan Chin Nam Centre for Parkinson's & Related Disorders, Faculty of Medicine, University of Malaya, Kuala Lumpur, Malaysia, **63** Department of Neurology, National Taiwan University Hospital, National Taiwan University College of Medicine, Taipei, Taiwan, **64** Pontificia Universidad Católica Argentina (UCA), Buenos Aires, Argentina, **65** Griffith Institute for Drug Discovery (GRIDD), School of Environment and Science, Griffith University, Brisbane, Queensland, Australia, **66** Sección Movimientos Anormales, Departamento de Neurociencias, Fleni, Buenos Aires, Argentina, **67** Argentine National Scientific and Technological Research Council (CONICET), Buenos Aires, Argentina, **68** Department of Genetics, Institute of Biology Roberto Alcantara Gomes, State University of Rio de Janeiro, Rio de Janeiro, Brazil, **69** Division of Neurology, Department of Medicine, Ramathibodi Hospital, Mahidol University, Rajthevi, Bangkok, Thailand, **70** Department of Neurology, Lund University, Lund, Sweden, **71** Department of Neurology, Skåne University Hospital, Lund, Sweden, **72** Tanz Centre for Research in Neurodegenerative Diseases, University of Toronto, Toronto, ON, Canada, **73** Medical Research Council Protein Phosphorylation and Ubiquitylation Unit, University of Dundee, Dundee, United Kingdom, **74** Molecular and Clinical Medicine, Ninewells Hospital and Medical School, University of Dundee, Dundee, United Kingdom, **75** Centre of Health Science, University of the Faroe Islands, Tórshavn, Faroe Islands, **76** Department of Occupational Medicine and Public Health, The Faroese Hospital System, Tórshavn, Faroe Islands, **77** Pavol Jozef Šafárik University in Košice, Košice, Slovakia, **78** Department of Neurology, University Hospital L. Pasteur, Kosice, Slovakia, **79** Neurology Service, State University of Rio de Janeiro, Rio de Janeiro, Rio de Janeiro, Brazil, **80** Department of Medicine, Medical Genetics and Pediatrics, University of Alberta, Edmonton, Alberta, Canada, **81** Behavioral and Movement Disorders Section, Ribeirão Preto Medical School, University of São Paulo, São Paulo, Brazil,

the LARGE PD study in Peru. Jean-Christophe Corvol received grants from Sanofi and the Michael J. Fox Foundation, consulting fees from Biogen, UCB, Denali, Idorsia, Prevail therapeutics, Theranexus, and honoraria from Biogen. Joana Damásio received honoraria from Zambon Pharmaceuticals. Anna De Rosa received grants for ROPAD – the Rostock International Parkinson's Disease Study, sponsored by Centogene and grants from Zambon and AIFA (Italian Agency of Drug), and she is member of the advisory board at BIAL. Joaquim Ferreira received grants from Fundação MSD (Portugal), Novartis, Medtronic, and Abbvie, and lecture fees from Lundbeck, Abbvie, BIAL, Biogen, Sunovion Pharmaceuticals, ONO, Affiris, and Zambon, payment for expert testimony from Novartis, and he participates in advisory boards of Lundbeck, Abbvie, BIAL, Affiris, Sunovion Pharmaceuticals, and Zambon; he is employed by CNS (Campus Néurologico Sénior) and the Medical Faculty of Lisbon. Emilia Gatto received consulting fees from Bago Argentina, honoraria from Bago Argentina,UCB, IPMDS, and Europharm, and participates in advisory boards of Bago Argentina and UCB. Nir Giladi received grants from the Michael J Fox Foundation, The National Parkinson Foundation, The European Union, The Israel Science Foundation Teva, NNE program, Biogen, Ionis, Sieratzki Family Foundation, and The Aufzien Academic Center in Tel-Aviv University; he holds royalties or licenses at Lysosomal Therapeutics (LTI); he received consulting fees from Sionara, NeuroDerm, Pharma2B, Denali, Neuron23, Sanofi-Genzyme, Biogen, and Abbvie; he received honoraria from Abbvie, Sanofi-Genzyme, and the Movement Disorder Society. Nobutaka Hattori received grants from the Japan Society for the Promotion of Science (JSPS; 18H04043, 21H04820, 19K22603), the Japan Agency for Medical Research and Development (AMED; JP20dm0307101, JP20dm0207070, JP20ek0109358, JP19ek0109393, JP19gm0710011, JP19km0405206), Health Labour Sciences Research Grant (20FC1049, H29-FC1-062, H29-FC1-033), and the Japan Science and Technology Agency (JPMJMS2024-5); he participates in advisory boards at Dai-Nippon Sumitomo Pharma, Takeda Pharmaceutical, Kyowa Kirin, TEIJIN PHARMA LIMITED, Novartis Pharma, Ono Pharmaceutical, Biogen Idec Japan, and Kissei Pharmaceutical; he receives consulting fees from Hisamitsu Pharma; he received honoraria from Dai-Nippon Sumitomo Pharma, Takeda Pharmaceutical, Kyowa Kirin, AbbVie GK, Nippon Boehringer Ingelheim, Otsuka Pharmaceutical, Novartis Pharma, Bristol-Myers Squibb, Ono Pharmaceutical, FP Pharmaceutical, Eisai, Kissei

**82** Department of Molecular Medicine, University of Pavia, Pavia, Italy, **83** Neurogenetics Research Centre, IRCCS Mondino Foundation, Pavia, Italy, **84** Department of Neurology, Donders Institute for Brain, Cognition, and Behavior, Radboud University Medical Centre, Nijmegen, The Netherlands, **85** Department of Neurology, Second Affiliated Hospital, College of Medicine, Zhejiang University, Hangzhou, Zhejiang, China, **86** Department of Neurology, Medical University, Vienna, Austria, **87** The Michael J. Fox Foundation for Parkinson's Research, New York, NY, United States of America

* christine.klein@neuro.uni-luebeck.de

## Abstract

Parkinson's disease (PD) is the fastest-growing neurodegenerative disorder, currently affecting ~7 million people worldwide. PD is clinically and genetically heterogeneous, with at least 10% of all cases explained by a monogenic cause or strong genetic risk factor. However, the vast majority of our present data on monogenic PD is based on the investigation of patients of European White ancestry, leaving a large knowledge gap on monogenic PD in underrepresented populations. Gene-targeted therapies are being developed at a fast pace and have started entering clinical trials. In light of these developments, building a global network of centers working on monogenic PD, fostering collaborative research, and establishing a clinical trial-ready cohort is imperative. Based on a systematic review of the English literature on monogenic PD and a successful team science approach, we have built up a network of 59 sites worldwide and have collected information on the availability of data, biomaterials, and facilities. To enable access to this resource and to foster collaboration across centers, as well as between academia and industry, we have developed an interactive map and online tool allowing for a quick overview of available resources, along with an option to filter for specific items of interest. This initiative is currently being merged with the Global Parkinson's Genetics Program (GP2), which will attract additional centers with a focus on underrepresented sites. This growing resource and tool will facilitate collaborative research and impact the development and testing of new therapies for monogenic and potentially for idiopathic PD patients.

## Introduction

As technologies for genetic testing are rapidly developing and are becoming more cost-effective, rare disorders and also rare subtypes of common disorders can increasingly be characterized genetically and thus stratified into genetically defined subgroups. This provides the basis not only to investigate disease mechanisms further but also to develop and test gene-targeted therapies. Parkinson's disease (PD), the most rapidly growing neurodegenerative disorder [1] can serve as a model disease for this approach, as PD is a common disorder affecting about 7 million people worldwide in 2022, including an estimated 700,000–840,000 (10–12%) hereditary cases with pathogenic genetic variants (monogenic PD and PD due to strong genetic risk factors). In light of the development of gene-targeted therapies for PD, identifying and characterizing carriers of specific genetic pathogenic variants is imperative.

Only a small fraction of the estimated number of subjects with monogenic PD worldwide are currently represented in the literature [2–5]. Most study centers publishing in the English language are located in Europe and North America [6]. There is evidence for variable

penetrance and expressivity of the disease across populations and ethnicities in monogenic forms of PD [2]. However, the majority of published cases are reported to be White and originate from Europe or North America, whereas most African, Asian, and South American populations, especially indigenous people, are highly underrepresented. Little is known about the actual worldwide epidemiology, frequency of population-specific variants, and possible local factors influencing the manifestation and course of the disease. Notably, even for published cases, the availability of detailed genetic and clinical data is limited, and genetically stratified clinical trial-ready cohorts are highly sought after.

Global collaboration both within academic research and with the pharmaceutical industry is paramount to expanding our knowledge of disease mechanisms, developing gene-targeted medications, and testing them in clinical trials. Building an international network of researchers from academia and industry will facilitate including more underrepresented populations in investigations and trials, both by enhancing the visibility of existing study centers and by providing genetic testing facilities. To open this network to new members and specifically allow researchers or clinicians from currently underrepresented countries to engage in these collaborative efforts, we established an online resource that provides an overview of centers worldwide working on monogenic PD and their facilities. While this effort sprung from the MJFF Global Genetic Parkinson's Disease Project [6], it has recently been integrated into the Global Parkinson's Genetics Program (GP2, https://gp2.org/), which will build global will and capacity while addressing emerging research questions in PD genetics [7].

## Methods

### How to find eligible centers?

Performing a systematic literature review on monogenic PD in the English literature [3, 4], our group identified the need for a new, systematic, and global approach to obtain more detailed clinical and genetic data from individuals with monogenic PD. Starting in March 2018, we contacted by email all corresponding authors from articles that we had included in the review and ran an online survey to collect information on the number of subjects with monogenic PD they follow, along with available clinical and genetic data. In a second step, we invited all researchers to provide individual case-level data and offered a small per-case fee to reimburse them for their effort to retrieve and enter the data in another online survey. The vast majority of participating centers expressed enthusiasm about our project, their strong interest in being part of the emerging study group, and support for further collaborative efforts.

To establish the online resource, we contacted all participating researchers, including their collaborators (n = 112). We received responses from 59 centers (53%) in 36 countries on all inhabited continents (Europe (n = 33), North (n = 4) and South America (n = 8), Asia (n = 10), Australia (n = 1), Africa (n = 3)). All centers have agreed to be contacted by researchers and the pharmaceutical industry, while the majority of centers (n = 41, 69%) would prefer not to be contacted directly but to have requests filtered first. The Monogenic Network of GP2 will handle collaboration requests. All participants provided written informed consent to the publication of the provided data. The ethics committee of the University of Lübeck confirmed that no ethical approval was required for this study.

### How to obtain the data of interest?

An essential part of this project was communication and information. Having identified eligible researchers, we sent out information emails about our project and kept all participants updated in regular progress emails. To further improve the retention rate, we ensured a quick

**Table 1. Items assessed in the online survey.**

| Site details | Name of site |
|---|---|
| | Principal investigator |
| | City/ Country |
| | Active in other PD projects |
| | New PD patients/year |
| | Genetic testing available |
| | How are genetic testing costs covered? |
| | Qualified genetic counseling available? |
| | Already performing clinical trials? |
| **Availability of clinical data** | Sex |
| | Age |
| | Age at onset |
| | UPDRS |
| | Hoehn and Yahr scale |
| | Nonmotor scales |
| | Olfactory testing |
| **Data sharing** | IRB consent |
| | Shipment outside the country |
| **Availability of biomaterials** | DNA/RNA/Serum/Plasma/whole blood/CSF/fibroblasts/iPSCs/brain tissue (storage?) |
| **Availability of Omics** | Genomics/ Transcriptomics/ Proteomics/ Metabolomics |
| **Laboratory facilities** | CSF handling |
| | Serum Plasma handling |
| | Culturing of fibroblasts |
| | Isolation of cells from whole blood |
| | Generation of iPSCs |
| | Neuropathology |
| | MRI |
| | SPECT/PET |
| | TCS |

PD: Parkinson's disease; UPDRS: Unified Parkinson's Disease Rating Scale; IRB: institutional review board; DNA: Deoxyribonucleic acid; RNA: Ribonucleic acid; CSF: cerebrospinal fluid; iPSCs: induced pluripotent stem cells; MRI: magnetic resonance imaging, SPECT/PET: Single-photon emission computed tomography/positron emission tomography, TCS: transcranial sonography

response in case of any queries and consistently named a specific contact person. When designing the survey, we aimed at providing an accessible online format, allowing different browser types and devices (again, our team was available and provided help in case of any technical difficulties), kept it brief, and phrased questions in a concise and explicit way (Table 1). Whenever possible, questions were designed to allow single-choice answers only, and free text fields were limited to maximize comparison between centers. For convenience, already available information (such as the name and location of the study center) was prefilled by our team. Participants received an invitation email including a link to their individual survey. Once complete, the survey was submitted by the participant. We sent out two rounds of reminder emails, personally addressing each participant, to improve the response rate and make sure all invitees received their email and got the opportunity to present their study center.

After completing the survey, we reviewed the data and double-checked for any inaccurate entries. In case of ambiguity, we reached out to the respective researcher for approval of the

correction. After this first round of quality checks, we programmed the online tool and uploaded the provided data. In a second round of quality checks and to allow for possible updates, we invited all participants to review and verify the provided data and obtained permission to publish the presented resource. Regular updates of the resource and map will be provided biannually, and ongoing projects and trials using the network will be listed on the website to facilitate further collaboration while avoiding duplication of efforts.

## Results

Our two-part online tool is now available at https://gp2networkdev.wpengine.com/monogenic-resource-map/. Firstly, it includes a searchable online table that allows for filtering for specific characteristics of interest (e.g., location of the center, available facilities, Table 1). Second, a map displays the locations of all research centers and provides individual profiles with information on each center (Fig 1).

The average number of new PD patients per year reported by the participants until May 2022 is 196. Genetic testing is regularly performed by 37 (63%) of the centers, and the costs are predominantly covered by in-house or external funding sources (39%), followed by coverage by health insurance and the patients themselves (10% each). The availability of biomaterials varies depending on how difficult they are to obtain, ranging from 53 centers (90%) reporting the availability of DNA for at least a subset of cases to 8 centers (14%) reporting the availability of brain tissue (Fig 2). Laboratory facilities that are part of the standard clinical workup like serum and plasma handling, are more frequently available (at 53 centers, 90%) than more specific ones that are more costly or require special training, such as the generation of induced pluripotent stem cells (at 24 centers, 41%; Fig 3).

## Discussion and conclusions

Using a powerful team science approach, we have established a global network of academic centers involved in the clinical care of patients with monogenic PD and with a high interest in collaboration, research, and gene-targeted therapies for PD. Based on a systematic search of

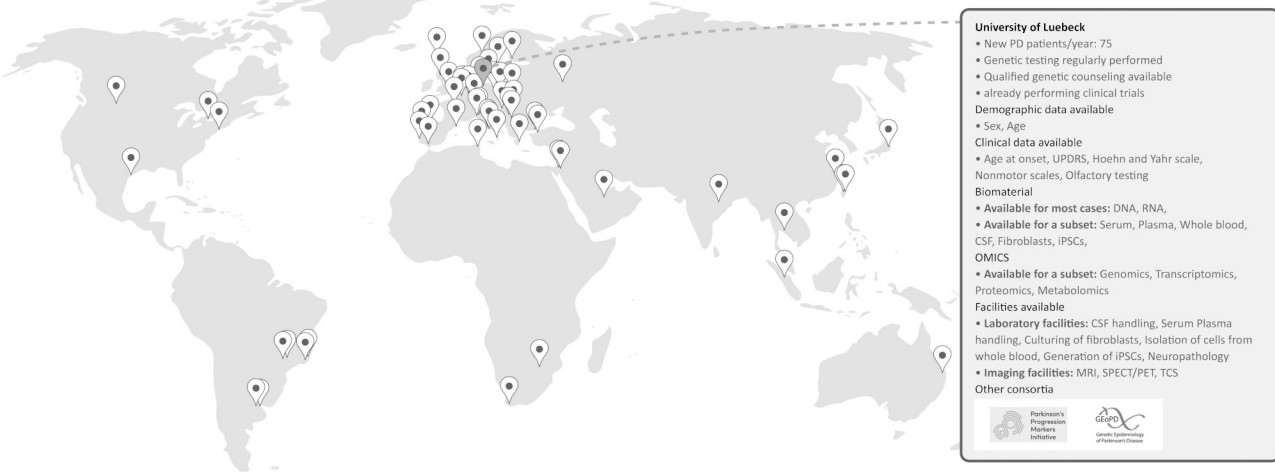

**Fig 1. The monogenic resource map.** This world map displays the distribution of all participating centers worldwide. The pins mark the location of the study centers. Clicking on a pin opens the respective study center profile. The profiles summarize information on the site and the availability of demographic and clinical data, biomaterials, methods, and facilities and provide information on other collaborative projects that the study center is part of.

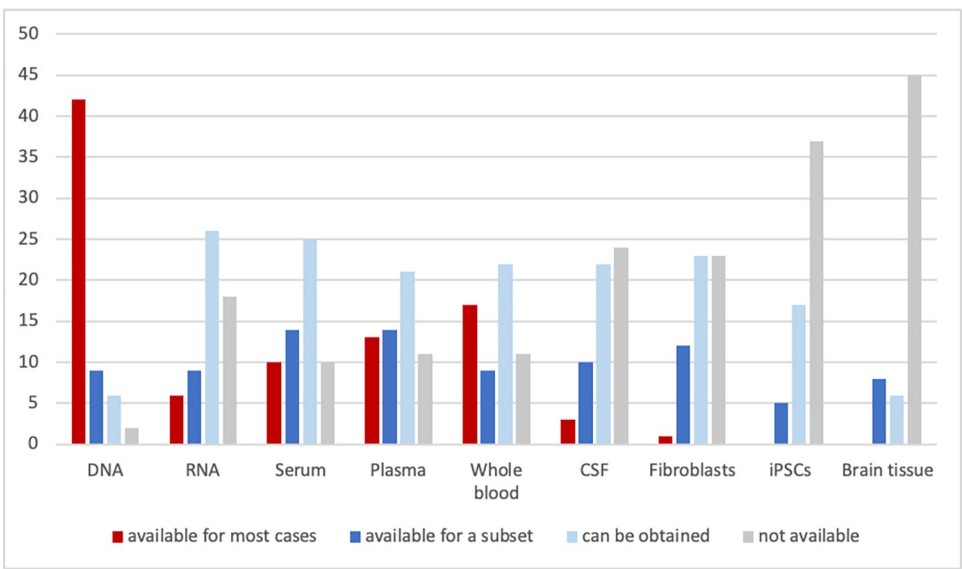

**Fig 2. Availability of biomaterials.** Number of centers reporting availability of the listed biomaterials. DNA: Deoxyribonucleic acid, RNA: Ribonucleic acid, CSF: cerebrospinal fluid, iPSCs: induced pluripotent stem cells.

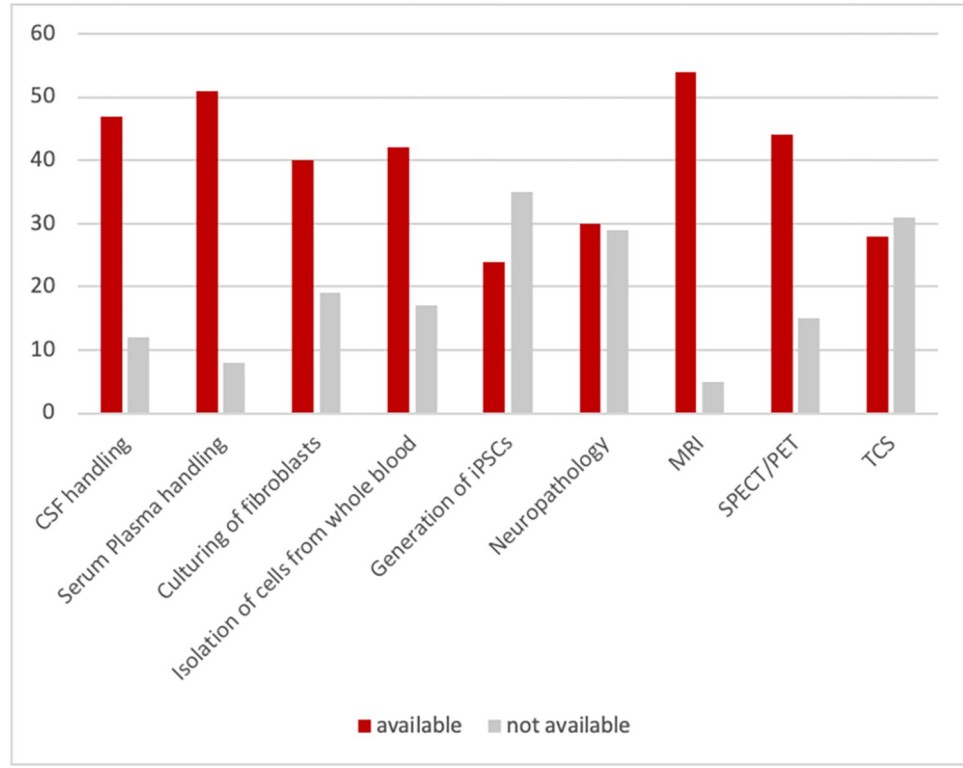

**Fig 3. Availability of methods and facilities.** Number of centers reporting availability of the listed methods and facilities. CSF: cerebrospinal fluid, iPSCs: induced pluripotent stem cells, MRI: magnetic resonance imaging, SPECT/PET: Single-photon emission computed tomography/positron emission tomography, TCS: transcranial sonography.

the world literature on monogenic PD published in the English language, we have tried to be as inclusive as possible. We have set up the project to serve as a growing resource open to all interested participants. Having focused our initial approach on articles in published in English is a limitation, as we certainly missed centers engaged in the care of patients with monogenic PD who did not publish on this topic or in a language other than English. This includes predominantly centers from regions with high percentages of underrepresented populations. In order to actively broaden this network and to reach out to active centers that have not yet published on monogenic PD (in English), the MJFF Global Genetic Parkinson's Disease Project is currently being merged within GP2, which presently includes 119 centers that will be approached to join this effort if not yet part of the network. Information on GP2 is already available in several languages ([https://gp2networkdev.wpengine.com/monogenic-resource-map/](https://gp2networkdev.wpengine.com/monogenic-resource-map/)), including Arabic, Mandarin, French, German, Japanese, and Spanish. Through this merger, we expect the resource to proliferate and, most importantly, to include an increasing number of previously underrepresented populations and centers.

To facilitate access to this resource and to foster collaboration across centers, as well as between academia and industry, we have developed an interactive map and online tool allowing for a quick overview and descriptive statistics on available data, materials, and facilities, as well as for filtering for specific items of interest, for example, centers who have induced pluripotent stem cell (iPSC) lines of *PRKN* mutation carriers.

We will exploit and expand existing networks through this project and resource to make more clinical and research centers visible and study subjects available to the clinical and basic PD research community. Furthermore, such studies will spur collaborations across global clinical sites and between academia and industry, thus facilitating and expediting the testing of new therapies in relevant PD genetic subtypes. Given the emerging knowledge about disease mechanisms contributing to PD pathogenesis, our hope is that different subpopulations of idiopathic PD subjects would also benefit from these large genetic cohort-based efforts.

## Acknowledgments

The authors would like to thank all contributing researchers for supporting this project.

## Author Contributions

**Conceptualization:** Eva-Juliane Vollstedt, Roger A. Barker, Dariusz Koziorowski, Shalini Padmanabhan, Christine Klein.

**Data curation:** Eva-Juliane Vollstedt, Harutyun Madoev, Jolanta Dorszewska.

**Formal analysis:** Eva-Juliane Vollstedt, Christine Klein.

**Funding acquisition:** Christine Klein.

**Investigation:** Eva-Juliane Vollstedt, Christine Klein.

**Methodology:** Eva-Juliane Vollstedt, Shalini Padmanabhan, Christine Klein.

**Project administration:** Eva-Juliane Vollstedt.

**Resources:** Eva-Juliane Vollstedt, Anna Aasly, Azlina Ahmad-Annuar, Bashayer Al-Mubarak, Roy N. Alcalay, Victoria Alvarez, Ignacio Amorin, Grazia Annesi, David Arkadir, Soraya Bardien, Roger A. Barker, Melinda Barkhuizen, A. Nazli Basak, Vincenzo Bonifati, Agnita Boon, Laura Brighina, Kathrin Brockmann, Andrea Carmine Belin, Jonathan Carr, Jordi Clarimon, Mario Cornejo-Olivas, Leonor Correia Guedes, Jean-Christophe Corvol, David Crosiers, Joana Damásio, Parimal Das, Patricia de Carvalho Aguiar, Anna De Rosa, Sibel

Ertan, Rosangela Ferese, Joaquim Ferreira, Emilia Gatto, Gençer Genç, Nir Giladi, Pilar Gómez-Garre, Hasmet Hanagasi, Nobutaka Hattori, Faycal Hentati, Dorota Hoffman-Zacharska, Sergey N. Illarioshkin, Joseph Jankovic, Silvia Jesús, Valtteri Kaasinen, Anneke Kievit, Peter Klivenyi, Vladimir Kostic, Dariusz Koziorowski, Andrea A. Kühn, Anthony E. Lang, Shen-Yang Lim, Chin-Hsien Lin, Katja Lohmann, Vladana Markovic, Mika Henrik Martikainen, George Mellick, Marcelo Merello, Lukasz Milanowski, Pablo Mir, Özgür Öztop-Çakmak, Márcia Mattos Gonçalves Pimentel, Teeratorn Pulkes, Andreas Puschmann, Ekaterina Rogaeva, Esther M. Sammler, Maria Skaalum Petersen, Matej Skorvanek, Mariana Spitz, Oksana Suchowersky, Ai Huey Tan, Pichet Termsarasab, Avner Thaler, Vitor Tumas, Enza Maria Valente, Bart van de Warrenburg, Caroline H. Williams-Gray, Ruey-Mei Wu, Baorong Zhang, Alexander Zimprich, Christine Klein.

**Software:** Eva-Juliane Vollstedt, Harutyun Madoev.

**Supervision:** Justin Solle, Shalini Padmanabhan, Christine Klein.

**Visualization:** Eva-Juliane Vollstedt.

**Writing – original draft:** Eva-Juliane Vollstedt, Christine Klein.

**Writing – review & editing:** Eva-Juliane Vollstedt, Harutyun Madoev, Anna Aasly, Azlina Ahmad-Annuar, Bashayer Al-Mubarak, Roy N. Alcalay, Victoria Alvarez, Ignacio Amorin, Grazia Annesi, David Arkadir, Soraya Bardien, Roger A. Barker, Melinda Barkhuizen, A. Nazli Basak, Vincenzo Bonifati, Agnita Boon, Laura Brighina, Kathrin Brockmann, Andrea Carmine Belin, Jonathan Carr, Jordi Clarimon, Mario Cornejo-Olivas, Leonor Correia Guedes, Jean-Christophe Corvol, David Crosiers, Joana Damásio, Parimal Das, Patricia de Carvalho Aguiar, Anna De Rosa, Jolanta Dorszewska, Sibel Ertan, Rosangela Ferese, Joaquim Ferreira, Emilia Gatto, Gençer Genç, Nir Giladi, Pilar Gómez-Garre, Hasmet Hanagasi, Nobutaka Hattori, Faycal Hentati, Dorota Hoffman-Zacharska, Sergey N. Illarioshkin, Joseph Jankovic, Silvia Jesús, Valtteri Kaasinen, Anneke Kievit, Peter Klivenyi, Vladimir Kostic, Dariusz Koziorowski, Andrea A. Kühn, Anthony E. Lang, Shen-Yang Lim, Chin-Hsien Lin, Katja Lohmann, Vladana Markovic, Mika Henrik Martikainen, George Mellick, Marcelo Merello, Lukasz Milanowski, Pablo Mir, Özgür Öztop-Çakmak, Márcia Mattos Gonçalves Pimentel, Teeratorn Pulkes, Andreas Puschmann, Ekaterina Rogaeva, Esther M. Sammler, Maria Skaalum Petersen, Matej Skorvanek, Mariana Spitz, Oksana Suchowersky, Ai Huey Tan, Pichet Termsarasab, Avner Thaler, Vitor Tumas, Enza Maria Valente, Bart van de Warrenburg, Caroline H. Williams-Gray, Ruey-Mei Wu, Baorong Zhang, Alexander Zimprich, Justin Solle, Shalini Padmanabhan, Christine Klein.

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
