## [Decision Letter · Decision Letter 0]

20 Mar 2023

PONE-D-22-35560Establishing an online resource to facilitate global collaboration and inclusion of underrepresented populations: experience from the MJFF Global Genetic Parkinson’s Disease ProjectPLOS ONE

Dear Dr. Klein,

Thank you for submitting your manuscript to PLOS ONE. After careful consideration, we feel that it has merit but does not fully meet PLOS ONE’s publication criteria as it currently stands. Therefore, we invite you to submit a revised version of the manuscript that addresses the points raised during the review process.

We look forward to receiving your revised manuscript.

Kind regards,

Ahmed Nasreldein, MD

Academic Editor

PLOS ONE

3. You indicated that ethical approval was not necessary for your study. We understand that the framework for ethical oversight requirements for studies of this type may differ depending on the setting and we would appreciate some further clarification regarding your research. Could you please provide further details on why your study is exempt from the need for approval and confirmation from your institutional review board or research ethics committee (e.g., in the form of a letter or email correspondence) that ethics review was not necessary for this study? Please include a copy of the correspondence as an "Other" file."

“CK received a grant for this project by the Michael J Fox Foundation (ID 15015.03, https://www.michaeljfox.org/).”

“This project is funded by the Michael J. Fox Foundation. Roger Barker and Caroline Williams-Gray are supported by the NIHR Cambridge Biomedical Research Centre (BRC-1215-20014).  The views expressed are those of the authors and not necessarily those of the NIHR or the Department of Health and Social Care.”

“CK received a grant for this project by the Michael J Fox Foundation (ID 15015.03, https://www.michaeljfox.org/).”

6. Thank you for stating the following in the Competing Interests section:

“I have read the journal's policy and the authors of this manuscript have the following competing interests: Roy N. Alcalay received consulting fees from Avrobio, Caraway, Ono Therapeutics, GSK, Merck, Sanofi, Janssen and grants from the Michael J. Fox Foundation, DOD, the Parkinson’s Disease Foundation, and the NIH. Melinda Barkhuizen received a PhD scholarship from the National Research Foundation of South Africa (grant numbers 89230 and 98217) and internal funding from the research center where the study was conducted (DST/NWU Preclinical Drug Development Platform, North-West University, South Africa); Ampath Pathology laboratories in South Africa donated services in the form of drawing participant blood for DNA extractions, Several neurologists in South Africa assisted the authors with identifying patients with Parkinson`s disease and referring them to the genotyping study; the North-West University, South Africa provided Ethical oversight and approval of the genotyping project. Vincenzo Bonifati received grants from Stichting Parkinson Fonds (Netherlands) and Alzheimer Nederland, he received honoraria from the International Parkinson and Movement Disorder Society (MDS), for lectures and as Chair of the International Congress Scientific Program Committee (2020-2021), and from Elsevier Ltd. as co-Editor in Chief of the journal Parkinsonism & Related Disorders (2018-current); he is unpaid member of the Stichting Parkinson Fonds (Netherlands). Kathrin Brockmann received grants from the German Federal Ministry of Education and Research (BMBF; PDStrat; FKZ 031L0137B) and from the German Center for Neurodegenerative Diseases (DZNE), consulting fees from Abbvie, Lundbeck, UCB, Zambon, Roche, and honoraria from Abbvie and UCB. Jordi Clarimon is full-time employee at Lundbeck A/S (Denmark). Mario Cornejo-Olivas has subcontracts with Cleveland Clinic and San Marcos Foundation for recruiting participants for the LARGE PD study in Peru. Jean-Christophe Corvol received grants from Sanofi and the Michael J. Fox Foundation, consulting fees from Biogen, UCB, Denali, Idorsia, Prevail therapeutics, Theranexus, and honoraria from Biogen. Joana Damásio received honoraria from Zambon Pharmaceuticals. Anna De Rosa received grants for ROPAD – the Rostock International Parkinson’s Disease Study, sponsored by Centogene and grants from Zambon and AIFA (Italian Agency of Drug), and she is member of the advisory board at BIAL. Joaquim Ferreira received grants from Fundação MSD (Portugal), Novartis, Medtronic, and Abbvie, and lecture fees from Lundbeck, Abbvie, BIAL, Biogen, Sunovion Pharmaceuticals, ONO, Affiris, and Zambon, payment for expert testimony from Novartis, and he participates in advisory boards of Lundbeck, Abbvie, BIAL, Affiris, Sunovion Pharmaceuticals, and Zambon; he is employed by CNS (Campus Néurologico Sénior) and the Medical Faculty of Lisbon. Emilia Gatto received consulting fees from Bago Argentina, honoraria from Bago Argentina,UCB, IPMDS, and Europharm, and participates in advisory boards of Bago Argentina and UCB. Nir Giladi received grants from the Michael J Fox Foundation, The National Parkinson Foundation, The European Union, The Israel Science Foundation Teva, NNE program, Biogen, Ionis, Sieratzki Family Foundation, and The Aufzien Academic Center in Tel-Aviv University; he holds royalties or licenses at Lysosomal Therapeutics (LTI); he received consulting fees from Sionara, NeuroDerm, Pharma2B, Denali, Neuron23, Sanofi-Genzyme, Biogen, and Abbvie; he received honoraria from Abbvie, Sanofi-Genzyme, and the Movement Disorder Society. Nobutaka Hattori received grants from the Japan Society for the Promotion of Science (JSPS; 18H04043, 21H04820, 19K22603), the Japan Agency for Medical Research and Development (AMED; JP20dm0307101, JP20dm0207070, JP20ek0109358, JP19ek0109393, JP19gm0710011, JP19km0405206), Health Labour Sciences Research Grant (20FC1049, H29-FC1-062, H29-FC1-033), and the Japan Science and Technology Agency (JPMJMS2024-5); he participates in advisory boards at Dai-Nippon Sumitomo Pharma, Takeda Pharmaceutical, Kyowa Kirin, TEIJIN PHARMA LIMITED, Novartis Pharma, Ono Pharmaceutical, Biogen Idec Japan, and Kissei Pharmaceutical; he receives consulting fees from Hisamitsu Pharma; he received honoraria from Dai-Nippon Sumitomo Pharma, Takeda Pharmaceutical, Kyowa Kirin, AbbVie GK, Nippon Boehringer Ingelheim, Otsuka Pharmaceutical, Novartis Pharma, Bristol-Myers Squibb, Ono Pharmaceutical, FP Pharmaceutical, Eisai, Kissei Pharmaceutical, Nihon Medi-physics, and Daiichi Sankyo; he received payment for expert testimony from Mitsubishi Tanabe Pharma; he received support for attending meeting from Takeda Pharmaceutical, and Kyowa Kirin; he has 17 patents planned or pending and 5 issued; he is Team Leader at the Neurodegenerative Disorders Collaborative Laboratory, RIKEN Center for Brain Science; he holds equity stock (8%) of PARKINSON Laboratories Co. Ltd.,  unrelated to the submitted work. Silvia Jesús has received grants from “Consejería de Salud y Familias” PI-0459-2018, “acción B Clínicos-Investigdores” B-0007-2019 and from the “instituto de Salud Carlos III” PI-18/01898; she received honoraria from Abbvie, Bial, Merz, UCB, Italfarmaco, Zambon and Server. Valtteri Kaasinen received grants from Turku University Foundation, Päivikki and Sakari Sohlberg Foundation, and the Finnish Cultural Foundation; he received consulting fees from Nordic Infucare, Abbvie, and

Adamant Health; he received honoraria from Nordic Infucare and Abbvie; he received Support for attending meetings and/or travel from Nordic Infucare; he participates on a Data Safety Monitoring Board or Advisory Board at Nordic Infucare and Abbvie; he is board member of the Finnish Neurological Society. Christine Klein received royalties from Oxford University Press; she received consulting fees from Centogene and Retromer Therapeutics; she received payment or honoraria from Desitin Pharma; she received support to attend meetings from the Movement Disorder Society; she participates on a Data Safety Monitoring Board or Advisory Board at the Else Kroener Fresenius Foundation. Peter Klivenyi received honoraria from Abbvie and Medtronic. Vladimir Kostic received Grant 17590 from the Ministry of Education, Science and Technological Development of Serbia; he received honoraria from Novartis and Boehringer Ingelheim. Shen-Yang Lim received the following grants, (i) Stipend - Global Parkinson's Genetics Program (GP2) Working Group Co-Lead Award, from the Michael J. Fox Foundation, (ii) Michael J. Fox Foundation (Grant 18305: "Identifying leucocyte and urine biomarkers in PD patients with LRRK2 G2385R variant"). George Mellick received grants from the National Health and Medical Research Foundation, Australia (APP1151854, APP1084560). Lukasz Milanowski received grants from the Polish National Agency for Academic Exchange Iwanowska’s Fellowship PPN/IWA/2018/1/00006/U/00001/01, the APDA, the Foundation for Polish Science (FNP) and the Haworth Family Professorship in Neurodegenerative Diseases Fund. Pablo mir received grants from the Spanish Ministry of Science and Innovation (RTC2019-007150-1), the Instituto de Salud Carlos III-Fondo Europeo de Desarrollo Regional (ISCIII-FEDER, PI19/01576), and the Consejería de Salud y Bienestar Social de la Junta de Andalucía (PE-0210-2018); he received payment or honoraria for lectures from Abbvie, Abbott, and Zambon; he received Support for attending meetings and travel from Abbvie. Shalini Padmanabhan is employed by the Michael J. Fox Foundation. Andreas Puschmann received grants from Region Skåne, Sweden, Skåne University Hospital, Sweden, The Swedish Parkinson Academy, The Swedish Parkinson Foundation (Parkinsonfonden), MultiPark – a strategic research environment at Lund University, and Bundy Academy, Sweden; he received honoraria from Elsevier (Associate editor for Parkinsonism and Related Disorders), from the International Parkinson and Movement Disorder Society (MDS), and from the International Association of Parkinsonism and Related Disorders; he received support for travels from the International Association of Parkinsonism and Related Disorders; he is board member of the International Association of Parkinsonism and Related Disorders. Mariana Spitz received funding from Centogene for the ROPAD study; she received support for travels from Roche. Justin Solle is employed by the Michel J. Fox Foundation. Oksana Suchowersky received grants from Roche and WaveLifeSciences; she holds royalties or licenses at UpToDate; she received consulting fees from Abbvie and Sunovion; she received honoraria from the World Parkinson Conference; she participates in advisory boards at Alexion; she is board member of the Parkinson Society Alberta. Pichet Termsarasab received book royalties from Elsevier and Springer Nature Switzerland; he received honoraria for manuscript writing from MedLink Neurology and for lectures from Viatris. Bart van de Warrenburg received grants from Radboudumc, ZonMW, Hersenstichting, Gossweiler Foundation, and the Michael J Fox Foundation; he holds royalties or licenses at BSL Springer Nature; he received consulting fees from uniQure; he received honoraria from UKM Medical Center Kuala Lumpur; he participates in Medical advisory board of patient organizations (unpaid); he is unpaid member of Steering committees or executive boards of various research consortia; he received a wearable sensor set by Brugling Fund/Hersenstichting. Caroline H. Williams-Gray collaborates with Astra-Zeneca on the microbiome in Parkinson’s disease; she received consultancy fees from Modus Outcomes and Evidera, Inc./GlaxoSmithKline; she received honoraria from Profile Pharma Limited. Eva-Juliane Vollstedt, Harutyun Madoev, Anna Aasly, Azlina Ahmad-Annuar, Bashayer Al-Mubarak, Victoria Alvarez, Ignacio Amorin, Grazia Annesi, David Arkadir, Soraya Bardien, Roger A. Barker, A. Nazli Basak, Agnita Boon, Laura Brighina, Andrea Carmine Belin, Jonathan Carr, Leonor Correia Guedes, David Crosiers, Parimal Das, Patricia de Carvalho Aguiar, Jolanta Dorszewska, Sibel Ertan, Rosangela Ferese, Gençer Genç, Pilar Gómez-Garre, Hasmet Hanagasi, Faycal Hentati, Dorota Hoffman-Zacharska, Sergey N. Illarioshkin, Joseph Jankovic,  Anneke Kievit, Dariusz Koziorowski, Andrea A. Kühn, Anthony E. Lang, Chin-Hsien Lin, Katja Lohmann, Vladana Markovic, Mika Henrik Martikainen, Marcelo Merello, Özgür Öztop-Çakmak, Márcia Mattos Gonçalves Pimentel, Teeratorn Pulkes, Ekaterina Rogaeva, Esther M. Sammler, Maria Skaalum Petersen, Matej Skorvanek, Ai Huey Tan, Avner Thaler, Vitor Tumas, Enza Maria Valente, Ruey-Mei Wu, Baorong Zhang, and Alexander Zimprich report no conflict of interest.”

7. PLOS requires an ORCID iD for the corresponding author in Editorial Manager on papers submitted after December 6th, 2016. Please ensure that you have an ORCID iD and that it is validated in Editorial Manager. To do this, go to ‘Update my Information’ (in the upper left-hand corner of the main menu), and click on the Fetch/Validate link next to the ORCID field. This will take you to the ORCID site and allow you to create a new iD or authenticate a pre-existing iD in Editorial Manager. Please see the following video for instructions on linking an ORCID iD to your Editorial Manager account: https://www.youtube.com/watch?v=_xcclfuvtxQ

8. We note that Figure 1 in your submission contain [map/satellite] images which may be copyrighted. All PLOS content is published under the Creative Commons Attribution License (CC BY 4.0), which means that the manuscript, images, and Supporting Information files will be freely available online, and any third party is permitted to access, download, copy, distribute, and use these materials in any way, even commercially, with proper attribution. For these reasons, we cannot publish previously copyrighted maps or satellite images created using proprietary data, such as Google software (Google Maps, Street View, and Earth). For more information, see our copyright guidelines: http://journals.plos.org/plosone/s/licenses-and-copyright.

Reviewers' comments:

Reviewer's Responses to Questions

**Comments to the Author**

1. Is the manuscript technically sound, and do the data support the conclusions?

Reviewer #1: Yes

Reviewer #2: Yes

2. Has the statistical analysis been performed appropriately and rigorously? 

Reviewer #1: Yes

Reviewer #2: Yes

3. Have the authors made all data underlying the findings in their manuscript fully available?

Reviewer #1: Yes

Reviewer #2: Yes

4. Is the manuscript presented in an intelligible fashion and written in standard English?

Reviewer #1: Yes

Reviewer #2: Yes

5. Review Comments to the Author

Reviewer #1: Thank you very much for the impressive work and congratulations on the new resources.

Here are some comments from my side:

Please check the link provided: https://gp2networkdev.wpengine.com/monogenic-resource-map/ as it doesn’t show the mentioned contents: the table and the map

Revise the Editing, some phrases are unclear (example: one sentence was highlighted in yellow in the attached manuscript)

Figures should be in a better resolution.

Methodology:

Is the mentioned systematic review done first to find the centres and researchers published elsewhere?

Would you please provide the flow chart of the conducted systematic review? or a table with the included studies upon which the whole process was base?

Reviewer #2: The paper is simple and outlines a methodology to advance research into monogenic Parkinson's disease.

My hope will be that advancing genetic research may be of value to small number of PD patients but the vast majority of PD patients in underrepresented populations need access to better care for their PD which is significantly lacking so there is effort to advance upcoming genetic treatment but minimal to no effort to advance existence treatments for millions of affected populations, perhaps reprioritization of funds and resources is required.

6. PLOS authors have the option to publish the peer review history of their article (what does this mean?). If published, this will include your full peer review and any attached files.

Reviewer #1: **Yes: **Shaimaa El-Jaafary

Reviewer #2: No

---

## [Author Response · Author response to Decision Letter 0]

27 May 2023

Dear Dr. Chenette and dear Dr. El-Jaafary,

We would like to thank you and the anonymous reviewer for the thoughtful review of our manuscript. We have tried to incorporate all of the helpful suggestions and comments as detailed below in our point-by-point reply.

- Thank you very much for providing the style templates, we have formatted the manuscript accordingly.

- For our study, we invited researchers worldwide to complete an online survey on available facilities and data, no actual patient data was shared. The researchers who participated were informed about our plan to publish the information they shared in our online resource tool and as a manuscript, and they all gave their consent for publication via email. 

3. You indicated that ethical approval was not necessary for your study. We understand that the framework for ethical oversight requirements for studies of this type may differ depending on the setting and we would appreciate some further clarification regarding your research. Could you please provide further details on why your study is exempt from the need for approval and confirmation from your institutional review board or research ethics committee (e.g., in the form of a letter or email correspondence) that ethics review was not necessary for this study? Please include a copy of the correspondence as an "Other" file."

- Our study was an online survey of PIs at various international centers dealing with genetic PD. We wanted to know what clinical or research facilities are available at the centers. For example, we asked whether there is a clinical study center, whether there is neuropathology on site, whether fibroblasts can be cultured, plasma extracted, etc. Thus, we are not interested in "direct" data from patients or subjects, but in a description of centers to facilitate clinical and research exchange between these centers dealing with rare genetic forms of PD. Attached please find a letter from our ethics committee confirming that no ethics vote was necessary for this study.

“CK received a grant for this project by the Michael J Fox Foundation (ID 15015.03, https://www.michaeljfox.org/).”

- The funders reviewed the study design and suggested additional items for the survey. They had no role in data collection and analysis, the decision to publish, or the preparation of the manuscript.

“This project is funded by the Michael J. Fox Foundation. Roger Barker and Caroline Williams-Gray are supported by the NIHR Cambridge Biomedical Research Centre (BRC-1215-20014). The views expressed are those of the authors and not necessarily those of the NIHR or the Department of Health and Social Care.”

“CK received a grant for this project by the Michael J Fox Foundation (ID 15015.03, https://www.michaeljfox.org/).”

- Thank you very much for pointing this out. We removed the funding-related text from the acknowledgment section and would like to ask you to amend the Funding Statement as follows: “CK received a grant for this project by the Michael J Fox Foundation (ID 15015.03, https://www.michaeljfox.org/). Roger Barker and Caroline Williams-Gray are supported by the NIHR Cambridge Biomedical Research Centre (BRC-1215-20014). The views expressed are those of the authors and not necessarily those of the NIHR or the Department of Health and Social Care.”

6. Thank you for stating the following in the Competing Interests section:

“I have read the journal's policy and the authors of this manuscript have the following competing interests: Roy N. Alcalay received consulting fees from Avrobio, Caraway, Ono Therapeutics, GSK, Merck, Sanofi, Janssen and grants from the Michael J. Fox Foundation, DOD, the Parkinson’s Disease Foundation, and the NIH. Melinda Barkhuizen received a PhD scholarship from the National Research Foundation of South Africa (grant numbers 89230 and 98217) and internal funding from the research center where the study was conducted (DST/NWU Preclinical Drug Development Platform, North-West University, South Africa); Ampath Pathology laboratories in South Africa donated services in the form of drawing participant blood for DNA extractions, Several neurologists in South Africa assisted the authors with identifying patients with Parkinson`s disease and referring them to the genotyping study; the North-West University, South Africa provided Ethical oversight and approval of the genotyping project. Vincenzo Bonifati received grants from Stichting Parkinson Fonds (Netherlands) and Alzheimer Nederland, he received honoraria from the International Parkinson and Movement Disorder Society (MDS), for lectures and as Chair of the International Congress Scientific Program Committee (2020-2021), and from Elsevier Ltd. as co-Editor in Chief of the journal Parkinsonism & Related Disorders (2018-current); he is unpaid member of the Stichting Parkinson Fonds (Netherlands). Kathrin Brockmann received grants from the German Federal Ministry of Education and Research (BMBF; PDStrat; FKZ 031L0137B) and from the German Center for Neurodegenerative Diseases (DZNE), consulting fees from Abbvie, Lundbeck, UCB, Zambon, Roche, and honoraria from Abbvie and UCB. Jordi Clarimon is full-time employee at Lundbeck A/S (Denmark). Mario Cornejo-Olivas has subcontracts with Cleveland Clinic and San Marcos Foundation for recruiting participants for the LARGE PD study in Peru. Jean-Christophe Corvol received grants from Sanofi and the Michael J. Fox Foundation, consulting fees from Biogen, UCB, Denali, Idorsia, Prevail therapeutics, Theranexus, and honoraria from Biogen. Joana Damásio received honoraria from Zambon Pharmaceuticals. Anna De Rosa received grants for ROPAD – the Rostock International Parkinson’s Disease Study, sponsored by Centogene and grants from Zambon and AIFA (Italian Agency of Drug), and she is member of the advisory board at BIAL. Joaquim Ferreira received grants from Fundação MSD (Portugal), Novartis, Medtronic, and Abbvie, and lecture fees from Lundbeck, Abbvie, BIAL, Biogen, Sunovion Pharmaceuticals, ONO, Affiris, and Zambon, payment for expert testimony from Novartis, and he participates in advisory boards of Lundbeck, Abbvie, BIAL, Affiris, Sunovion Pharmaceuticals, and Zambon; he is employed by CNS (Campus Néurologico Sénior) and the Medical Faculty of Lisbon. Emilia Gatto received consulting fees from Bago Argentina, honoraria from Bago Argentina,UCB, IPMDS, and Europharm, and participates in advisory boards of Bago Argentina and UCB. Nir Giladi received grants from the Michael J Fox Foundation, The National Parkinson Foundation, The European Union, The Israel Science Foundation Teva, NNE program, Biogen, Ionis, Sieratzki Family Foundation, and The Aufzien Academic Center in Tel-Aviv University; he holds royalties or licenses at Lysosomal Therapeutics (LTI); he received consulting fees from Sionara, NeuroDerm, Pharma2B, Denali, Neuron23, Sanofi-Genzyme, Biogen, and Abbvie; he received honoraria from Abbvie, Sanofi-Genzyme, and the Movement Disorder Society. Nobutaka Hattori received grants from the Japan Society for the Promotion of Science (JSPS; 18H04043, 21H04820, 19K22603), the Japan Agency for Medical Research and Development (AMED; JP20dm0307101, JP20dm0207070, JP20ek0109358, JP19ek0109393, JP19gm0710011, JP19km0405206), Health Labour Sciences Research Grant (20FC1049, H29-FC1-062, H29-FC1-033), and the Japan Science and Technology Agency (JPMJMS2024-5); he participates in advisory boards at Dai-Nippon Sumitomo Pharma, Takeda Pharmaceutical, Kyowa Kirin, TEIJIN PHARMA LIMITED, Novartis Pharma, Ono Pharmaceutical, Biogen Idec Japan, and Kissei Pharmaceutical; he receives consulting fees from Hisamitsu Pharma; he received honoraria from Dai-Nippon Sumitomo Pharma, Takeda Pharmaceutical, Kyowa Kirin, AbbVie GK, Nippon Boehringer Ingelheim, Otsuka Pharmaceutical, Novartis Pharma, Bristol-Myers Squibb, Ono Pharmaceutical, FP Pharmaceutical, Eisai, Kissei Pharmaceutical, Nihon Medi-physics, and Daiichi Sankyo; he received payment for expert testimony from Mitsubishi Tanabe Pharma; he received support for attending meeting from Takeda Pharmaceutical, and Kyowa Kirin; he has 17 patents planned or pending and 5 issued; he is Team Leader at the Neurodegenerative Disorders Collaborative Laboratory, RIKEN Center for Brain Science; he holds equity stock (8%) of PARKINSON Laboratories Co. Ltd., unrelated to the submitted work. Silvia Jesús has received grants from “Consejería de Salud y Familias” PI-0459-2018, “acción B Clínicos-Investigdores” B-0007-2019 and from the “instituto de Salud Carlos III” PI-18/01898; she received honoraria from Abbvie, Bial, Merz, UCB, Italfarmaco, Zambon and Server. Valtteri Kaasinen received grants from Turku University Foundation, Päivikki and Sakari Sohlberg Foundation, and the Finnish Cultural Foundation; he received consulting fees from Nordic Infucare, Abbvie, and Adamant Health; he received honoraria from Nordic Infucare and Abbvie; he received Support for attending meetings and/or travel from Nordic Infucare; he participates on a Data Safety Monitoring Board or Advisory Board at Nordic Infucare and Abbvie; he is board member of the Finnish Neurological Society. Christine Klein received royalties from Oxford University Press; she received consulting fees from Centogene and Retromer Therapeutics; she received payment or honoraria from Desitin Pharma; she received support to attend meetings from the Movement Disorder Society; she participates on a Data Safety Monitoring Board or Advisory Board at the Else Kroener Fresenius Foundation. Peter Klivenyi received honoraria from Abbvie and Medtronic. Vladimir Kostic received Grant 17590 from the Ministry of Education, Science and Technological Development of Serbia; he received honoraria from Novartis and Boehringer Ingelheim. Shen-Yang Lim received the following grants, (i) Stipend - Global Parkinson's Genetics Program (GP2) Working Group Co-Lead Award, from the Michael J. Fox Foundation, (ii) Michael J. Fox Foundation (Grant 18305: "Identifying leucocyte and urine biomarkers in PD patients with LRRK2 G2385R variant"). George Mellick received grants from the National Health and Medical Research Foundation, Australia (APP1151854, APP1084560). Lukasz Milanowski received grants from the Polish National Agency for Academic Exchange Iwanowska’s Fellowship PPN/IWA/2018/1/00006/U/00001/01, the APDA, the Foundation for Polish Science (FNP) and the Haworth Family Professorship in Neurodegenerative Diseases Fund. Pablo mir received grants from the Spanish Ministry of Science and Innovation (RTC2019-007150-1), the Instituto de Salud Carlos III-Fondo Europeo de Desarrollo Regional (ISCIII-FEDER, PI19/01576), and the Consejería de Salud y Bienestar Social de la Junta de Andalucía (PE-0210-2018); he received payment or honoraria for lectures from Abbvie, Abbott, and Zambon; he received Support for attending meetings and travel from Abbvie. Shalini Padmanabhan is employed by the Michael J. Fox Foundation. Andreas Puschmann received grants from Region Skåne, Sweden, Skåne University Hospital, Sweden, The Swedish Parkinson Academy, The Swedish Parkinson Foundation (Parkinsonfonden), MultiPark – a strategic research environment at Lund University, and Bundy Academy, Sweden; he received honoraria from Elsevier (Associate editor for Parkinsonism and Related Disorders), from the International Parkinson and Movement Disorder Society (MDS), and from the International Association of Parkinsonism and Related Disorders; he received support for travels from the International Association of Parkinsonism and Related Disorders; he is board member of the International Association of Parkinsonism and Related Disorders. Mariana Spitz received funding from Centogene for the ROPAD study; she received support for travels from Roche. Justin Solle is employed by the Michel J. Fox Foundation. Oksana Suchowersky received grants from Roche and WaveLifeSciences; she holds royalties or licenses at UpToDate; she received consulting fees from Abbvie and Sunovion; she received honoraria from the World Parkinson Conference; she participates in advisory boards at Alexion; she is board member of the Parkinson Society Alberta. Pichet Termsarasab received book royalties from Elsevier and Springer Nature Switzerland; he received honoraria for manuscript writing from MedLink Neurology and for lectures from Viatris. Bart van de Warrenburg received grants from Radboudumc, ZonMW, Hersenstichting, Gossweiler Foundation, and the Michael J Fox Foundation; he holds royalties or licenses at BSL Springer Nature; he received consulting fees from uniQure; he received honoraria from UKM Medical Center Kuala Lumpur; he participates in Medical advisory board of patient organizations (unpaid); he is unpaid member of Steering committees or executive boards of various research consortia; he received a wearable sensor set by Brugling Fund/Hersenstichting. Caroline H. Williams-Gray collaborates with Astra-Zeneca on the microbiome in Parkinson’s disease; she received consultancy fees from Modus Outcomes and Evidera, Inc./GlaxoSmithKline; she received honoraria from Profile Pharma Limited. Eva-Juliane Vollstedt, Harutyun Madoev, Anna Aasly, Azlina Ahmad-Annuar, Bashayer Al-Mubarak, Victoria Alvarez, Ignacio Amorin, Grazia Annesi, David Arkadir, Soraya Bardien, Roger A. Barker, A. Nazli Basak, Agnita Boon, Laura Brighina, Andrea Carmine Belin, Jonathan Carr, Leonor Correia Guedes, David Crosiers, Parimal Das, Patricia de Carvalho Aguiar, Jolanta Dorszewska, Sibel Ertan, Rosangela Ferese, Gençer Genç, Pilar Gómez-Garre, Hasmet Hanagasi, Faycal Hentati, Dorota Hoffman-Zacharska, Sergey N. Illarioshkin, Joseph Jankovic, Anneke Kievit, Dariusz Koziorowski, Andrea A. Kühn, Anthony E. Lang, Chin-Hsien Lin, Katja Lohmann, Vladana Markovic, Mika Henrik Martikainen, Marcelo Merello, Özgür Öztop-Çakmak, Márcia Mattos Gonçalves Pimentel, Teeratorn Pulkes, Ekaterina Rogaeva, Esther M. Sammler, Maria Skaalum Petersen, Matej Skorvanek, Ai Huey Tan, Avner Thaler, Vitor Tumas, Enza Maria Valente, Ruey-Mei Wu, Baorong Zhang, and Alexander Zimprich report no conflict of interest.”

- We confirm that the reported competing interest section does not alter our adherence to the PLOS ONE policies od sharing data and materials. Please include the following Competing Interest statement: "This does not alter our adherence to PLOS ONE policies on sharing data and materials.”

7. PLOS requires an ORCID iD for the corresponding author in Editorial Manager on papers submitted after December 6th, 2016. Please ensure that you have an ORCID iD and that it is validated in Editorial Manager. To do this, go to ‘Update my Information’ (in the upper left-hand corner of the main menu), and click on the Fetch/Validate link next to the ORCID field. This will take you to the ORCID site and allow you to create a new iD or authenticate a pre-existing iD in Editorial Manager. Please see the following video for instructions on linking an ORCID iD to your Editorial Manager account: https://www.youtube.com/watch?v=_xcclfuvtxQ

- We added the ORCID iD for the corresponding author as requested. 

8. We note that Figure 1 in your submission contain [map/satellite] images which may be copyrighted. All PLOS content is published under the Creative Commons Attribution License (CC BY 4.0), which means that the manuscript, images, and Supporting Information files will be freely available online, and any third party is permitted to access, download, copy, distribute, and use these materials in any way, even commercially, with proper attribution. For these reasons, we cannot publish previously copyrighted maps or satellite images created using proprietary data, such as Google software (Google Maps, Street View, and Earth). For more information, see our copyright guidelines: http://journals.plos.org/plosone/s/licenses-and-copyright.

- Thank you very much for your advice regarding the copyright. We altered Figure 1 using a non-copyrighted graphic. 

- We confirm that the provided reference list is correct and nine of the cited papers have been retracted.

Review Comments to the Author

Reviewer #1: 

Thank you very much for the impressive work and congratulations on the new resources.

Here are some comments from my side:

Please check the link provided: https://gp2networkdev.wpengine.com/monogenic-resource-map/ as it doesn’t show the mentioned contents: the table and the map

- The link is now fully functional in all common browsers (Safari, Firefox, Google Chrome, and Microsoft Edge).

Revise the Editing, some phrases are unclear (example: one sentence was highlighted in yellow in the attached manuscript)

- Thank you very much for pointing this out, we corrected the respective sentence.

Figures should be in a better resolution.

- We now provided all figures in a higher resolution.

Methodology:

Is the mentioned systematic review done first to find the centres and researchers published elsewhere?

Would you please provide the flow chart of the conducted systematic review? or a table with the included studies upon which the whole process was base?

- The identification of suitable study centers and researchers was indeed part of two other projects conducted prior to this study by our team (MDSGene, PMID 29644727 and 30357936; MJFF Global Genetic PD Project PMID 31155756 and 36692014). A flow chart on the systematic review is included as Figure 2 in “Using global team science to identify genetic Parkinson’s disease worldwide”, Vollstedt et al. 2019 (PMID 31155756, please see figure below). 

 

Figure 2 from “Using global team science to identify genetic Parkinson’s disease worldwide”, Vollstedt et al (PMID 31155756): 

Response quantification.

Of the n=336 researchers we contacted, n=125 (98% of those who completed the survey) were interested in further collaboration on genetic PD. Of those researchers who did not respond to our invitation to participate, we identified n=52 (29%) to be affiliated with one of the responding centers.

Reviewer #2: 

The paper is simple and outlines a methodology to advance research into monogenic Parkinson's disease.

My hope will be that advancing genetic research may be of value to small number of PD patients but the vast majority of PD patients in underrepresented populations need access to better care for their PD which is significantly lacking so there is effort to advance upcoming genetic treatment but minimal to no effort to advance existence treatments for millions of affected populations, perhaps reprioritization of funds and resources is required.

- Thank you very much for sharing your thoughts on this very important matter. We also think that there is an urgent need to improve patient care for individuals with idiopathic PD. Our hope is that studying disease mechanisms in genetically predefined groups of patients will provide valuable insights into the pathophysiology of the disease overall and will facilitate drug development and improve patient care for individuals with idiopathic PD as well. The disparity the reviewer mentions is a major unmet need. With the present manuscript and resource, we hope to highlight currently underrepresented centers and their resources (which are often unknown to other researchers or to pharmaceutical companies). Some of these underrepresented sites follow a relatively large number of pathogenic variant carriers; thus, our map may also aid in flagging such centers that would be good candidates for clinical trial sites and, thus, could offer their patients inclusion in gene-targeted clinical trials.

Thank you very much for your consideration.

Christine Klein, MD, FEAN

---

## [Decision Letter · Decision Letter 1]

29 Aug 2023

PONE-D-22-35560R1Establishing an online resource to facilitate global collaboration and inclusion of underrepresented populations: experience from the MJFF Global Genetic Parkinson’s Disease ProjectPLOS ONE

Dear Dr. Klein,

Thank you for submitting your manuscript to PLOS ONE. After careful consideration, we feel that it has merit but does not fully meet PLOS ONE’s publication criteria as it currently stands. Therefore, we invite you to submit a revised version of the manuscript that addresses the points raised during the review process.

We look forward to receiving your revised manuscript.

Kind regards,

Amina Nasri

Academic Editor

PLOS ONE

Journal Requirements:

Reviewers' comments:

Reviewer's Responses to Questions

**Comments to the Author**

1. If the authors have adequately addressed your comments raised in a previous round of review and you feel that this manuscript is now acceptable for publication, you may indicate that here to bypass the “Comments to the Author” section, enter your conflict of interest statement in the “Confidential to Editor” section, and submit your "Accept" recommendation.

Reviewer #1: All comments have been addressed

Reviewer #3: (No Response)

2. Is the manuscript technically sound, and do the data support the conclusions?

Reviewer #1: Yes

Reviewer #3: Yes

3. Has the statistical analysis been performed appropriately and rigorously? 

Reviewer #1: Yes

Reviewer #3: N/A

4. Have the authors made all data underlying the findings in their manuscript fully available?

Reviewer #1: Yes

Reviewer #3: Yes

5. Is the manuscript presented in an intelligible fashion and written in standard English?

Reviewer #1: Yes

Reviewer #3: Yes

6. Review Comments to the Author

Reviewer #1: Dear Dr. Klein and co-authers.

Thank you very much for addressing all the comments.

The revised version only contain fig.2 from Vollstedt et al. So, please provide the figures (number 1,2,3) mentioned within the text in higher resolution as modified.

Reviewer #3: The authors have addressed the comments of the reviewers in detail and done so satisfactorily. The overall objective of the project described is to increase representation of under-represented populations. This flags one of the apparent shortcomings of the project which is that although a very significant population of URPs is in sub-Saharan black African populations across east, west and central Africa, the sites included in the efforts starkly exclude any sites from these African regions. Northern Africa (Tunisia) and Southern Africa are not typically regarded as underrepresented in terms of the ancestry of the population and their inclusion in genetic studies and one would presume that it would be important to have included other African regions. Indeed, excluding a significant underrepresented population in a project that seeks to enhance inclusion of underrepresented populations does seem counter-intuitive. The authors may wish to either clearly describe this as a limitation or, in implementing the initiative, address this drawback.

7. PLOS authors have the option to publish the peer review history of their article (what does this mean?). If published, this will include your full peer review and any attached files.

Reviewer #1: **Yes: **Shaimaa El-Jaafary

Reviewer #3: No

---

## [Author Response · Author response to Decision Letter 1]

5 Sep 2023

Dear Dr. Chenette,

We would like to thank you, Dr. El-Jaafary, and the anonymous Reviewer #3 again for the second review of our manuscript. We have addressed the reviewer’s comment below and amended the manuscript accordingly.

Reviewer #3: The authors have addressed the comments of the reviewers in detail and done so satisfactorily. The overall objective of the project described is to increase representation of under-represented populations. This flags one of the apparent shortcomings of the project which is that although a very significant population of URPs is in sub-Saharan black African populations across east, west and central Africa, the sites included in the efforts starkly exclude any sites from these African regions. Northern Africa (Tunisia) and Southern Africa are not typically regarded as underrepresented in terms of the ancestry of the population and their inclusion in genetic studies and one would presume that it would be important to have included other African regions. Indeed, excluding a significant underrepresented population in a project that seeks to enhance inclusion of underrepresented populations does seem counter-intuitive. The authors may wish to either clearly describe this as a limitation or, in implementing the initiative, address this drawback.

Thank you very much for alluding to this misunderstanding. The reason for missing these centers is actually not active exclusion but lies in our initial approach to identifying suitable centers. For the initial step of this project, we contacted the corresponding authors of publications on genetic Parkinson’s disease that we had systematically identified for the MDSGene project (www.mdsgene.org). MDSGene is a database including individual-level data on subjects with genetic causes for movement disorders that have been extracted from publications available in English. Unfortunately, with this approach, we likely missed centers that could have been included, as they may have published in languages other than English. 

We strongly agree that the lack of centers with underrepresented populations is currently a limitation of this project. In order to further expand the network and to actively include underrepresented populations, this project is currently being integrated into the Global Parkinson’s Genetic Program (GP2). GP2 specifically focuses on the inclusion of previously underrepresented populations and will consequently make more clinical and research centers visible and global study participants available to the clinical and basic PD research community. We have highlighted this limitation and outlook in the discussion section of our manuscript accordingly.

Thank you very much for your consideration.

Christine Klein, MD, FEAN

---

## [Editor Report · Decision Letter 2]

14 Sep 2023

Establishing an online resource to facilitate global collaboration and inclusion of underrepresented populations: experience from the MJFF Global Genetic Parkinson’s Disease Project

PONE-D-22-35560R2

Dear Dr. Christine Klein,

We’re pleased to inform you that your manuscript has been judged scientifically suitable for publication and will be formally accepted for publication once it meets all outstanding technical requirements.

Kind regards,

Amina Nasri

Academic Editor

PLOS ONE
---

## [Editor Report · Acceptance letter]

25 Sep 2023

PONE-D-22-35560R2 

Establishing an online resource to facilitate global collaboration and inclusion of underrepresented populations: experience from the MJFF Global Genetic Parkinson’s Disease Project 

Dear Dr. Klein:

I'm pleased to inform you that your manuscript has been deemed suitable for publication in PLOS ONE. Congratulations! Your manuscript is now with our production department. 

Kind regards, 

on behalf of

Dr. Amina Nasri 

Academic Editor

PLOS ONE